# THC-Reduced *Cannabis sativa* L.—How Does the Solvent Determine the Bioavailability of Cannabinoids Given Orally?

**DOI:** 10.3390/nu15122646

**Published:** 2023-06-06

**Authors:** Joanna Bartkowiak-Wieczorek, Edyta Mądry, Michał Książkiewicz, Jakub Winkler-Galicki, Milena Szalata, Marlena Szalata, Ulises Elizalde Jiménez, Karolina Wielgus, Edmund Grześkowiak, Ryszard Słomski, Agnieszka Bienert

**Affiliations:** 1Physiology Department, Poznan University of Medical Sciences, ul. Święcickiego 6, 61-861 Poznań, Poland; emadry@ump.edu.pl (E.M.); jwinklergalicki@ump.edu.pl (J.W.-G.); 2Cannabitey s.c. Poznań, ul. Uniwersytetu Poznańskiego 10/B123, 61-614 Poznań, Poland; michal.ksiazkiewicz@gmail.com; 3Department of Biotechnology, Institute of Natural Fibres and Medicinal Plants National Research Institute, Wojska Polskiego 71B, 60-630 Poznań, Poland; milena.szalata@iwnirz.pl (M.S.); slomski@up.poznan.pl (R.S.); 4Department of Biochemistry and Biotechnology, Poznań University of Life Sciences, ul. Dojazd 11, 60-632 Poznań, Poland; szalata@up.poznan.pl; 5Neuromed Consultorios, José Ibarra Olivares 106, Centro, Pachuca de Soto 42000, Hidalgo, Mexico; ulises.elizalde.j96@gmail.com; 6Centro Médico Privado Sanatorio Ortega, José Ibarra Olivares 105, Centro, Pachuca de Soto 42000, Hidalgo, Mexico; 7Department of Pediatric Gastroenterology and Metabolic Diseases, Poznan University of Medical Sciences, Szpitalna Street 27/33, 60-572 Poznań, Poland; kwielgus@ump.edu.pl; 8Department of Clinical Pharmacy and Biopharmacy, Poznan University of Medical Sciences, ul. Rokietnicka 3, 60-806 Poznań, Poland; grzesko@ump.edu.pl (E.G.); agbienert@ump.edu.pl (A.B.)

**Keywords:** endocannabinoid system, cannabidiol (CBD), tetrahydrocannabinol (THC), bioavailability, pharmacokinetics

## Abstract

The bioavailability levels of cannabidiol (CBD) and tetrahydrocannabinol (THC) determine their pharmacological effects. Therefore, for medical purposes, it is essential to obtain extracts containing the lowest possible content of the psychogenic component THC. In our extract, the CBD/THC ratio was 16:1, which is a high level compared to available medical preparations, where it is, on average, 1:1. This study assessed the bioavailability and stability of CBD and THC derived from *Cannabis sativa* L. with reduced THC content. The extract was orally administered (30 mg/kg) in two solvents, Rapae oleum and Cremophor, to forty-eight Wistar rats. The whole-blood and brain concentrations of CBD and THC were measured using liquid chromatography coupled with mass spectrometry detection. Much higher concentrations of CBD than THC were observed for both solvents in the whole-blood and brain after oral administration of the *Cannabis sativa* extract with a decreased THC content. The total bioavailability of both CBD and THC was higher for Rapae oleum compared to Cremophor. Some of the CBD was converted into THC in the body, which should be considered when using *Cannabis sativa* for medical purposes. The THC-reduced hemp extract in this study is a promising candidate for medical applications.

## 1. Introduction

The endocannabinoid system maintains physiological homeostasis, consisting of CB1 and CB2 receptors, their endogenous agonists, and enzymes that regulate their synthesis and degradation. CB1 receptors are mostly found in the brain, while CB2 receptors are expressed on the surface of lymphocytes. The endocannabinoid system regulates energy metabolism, the sensation of pain, immunological/inflammatory processes and motor activity, as well as affecting mood, motivation, memory and food supply [1].

The popularity of hemp and its therapeutical potential has been known since before the time of Christ [2]. The numerous benefits of hemp administration for humans and animals are due to their rich supply of biologically active compounds, among which Δ-9-tetrahydrocannabinol (THC) and cannabidiol (CBD) are assigned the largest roles [3]. *Cannabis* can be administrated in various forms, such as by smoking, vaporisation, taking infusions, tinctures, oils and pastes, as well as being applied directly to the skin or taken in the form of food. Each of these routes is characterised by a different therapeutic effect, one resulting from the variable bioavailability and pharmacokinetics of the cannabinoids [4].

Preclinical animal studies have provided evidence of the beneficial effects of cannabinoids from *Cannabis sativa* on cardiovascular disorders, cancer treatment, pain therapy, respiratory diseases and metabolic disorders, promoting further research [5] confirming their positive therapeutic effects in humans [6]. Of the two main active compounds of hemp, THC is responsible for more of the plant’s psychoactive properties. The bioavailability levels of THC, CBD and their derivatives depends on the route of administration of these compounds. Oral administration directs these substances into the bloodstream via the portal circulatory system. After their absorption, they go to the liver (first-pass effect), where they are metabolized with the participation of liver enzymes (CYP3A4, CYP2D6, CYP2C9, CYP1A2, CYP2B6, and CYP2C19) [7]. 

THC is converted into the psychoactive compound 11-COOH-THC and then into 11-OH-THC, which has no psychoactive activity. The THC metabolite 11-COOH-THC is the major glucuronide conjugate in the urine, while the THC metabolite 11-OH-THC is the dominant one in the faeces [8]. CBD is hydroxylated to 7-COOH derivatives, which are excreted unchanged or as glucuronide conjugates. The conversion of CBD to the psychotropic forms Δ9-THC and Δ8-THC under the influence of hydrochloric acid has been proven beyond doubt in many studies, and it is still the subject of intensive studies. In vitro studies have demonstrated that under acidic conditions, CBD is converted to ∆9-THC, as well as ∆7-THC, ∆8-THC, ∆10-THC, ∆11-THC and iso-THC [9]. Therefore, it is crucial to understand the bioavailability and stability of CBD and THC after oral administration.

The lipophilic nature of the compounds contained in *Cannabis sativa* contributes to their low solubility in water [10]; therefore, the search is ongoing for solvents that not only contribute to better absorption and distribution of cannabinoids but, importantly, improve their total bioavailability. Typically, oil compounds are commonly used as carriers for most lipophilic drugs [11]. It is suggested that non-ionic compounds such as Cremophor, which have hydrophilic properties, may emulsify and dissolve lipophilic molecules by forming micelles to enclose the lipophilic molecules [12]. 

In order to perform the research, an extract of *Cannabis sativa*, referred to as “THC-reduced hemp extract” was used. This means that the plant *Cannabis sativa* L., and any part of that plant or any compound, manufacture, salt, derivative, mixture, preparation, resin, or oil of that plant contains a reduced percentage of THC in relation to CBD (CBD:THC = 16/1; compared to drugs available on the medical market (e.g., dronabinol, nabilone, and Sativex—CBD:THC = 1/1 [13]) which are characterized by a high content of THC).

The purpose of this study was to assess the stability and bioavailability of CBD and THC from a *Cannabis sativa* extract with a reduced content of THC, as administrated orally in two different solvents. 

## 2. Methods

### 2.1. Characteristics of the Hemp Extract with a Reduced THC Content

The hemp variety KC Dora, a Hungarian variety of fibrous monoecious hemp bred at Agromag Kft, was used in this study, with the following extract composition:**Component****CBD-A****CBD****Δ9-THC-A****Δ9-THC**Concentration [mg/g]1.2215.20.1513.3

This extract with the described composition was used to prepare preparations in both solvents, namely, Rapae oleum and in a mixture of Cremophor/ethyl alcohol 96%/NaCl 0.9%, in the ratio 1:1:18.

### 2.2. Hemp Extract Preparation

The hemp extract was prepared using panicles of the Hungarian monoecious hemp variety KC Dora which was cultivated in experimental plots (Institute of Natural Fibres & Medicinal Plants, Poznan, Poland). The cultivation conditions included nitrogen fertilisation (30 kg/ha) and a seed-sowing density of 30 kg of seeds/ha. The panicles were harvested at the late-flowering stage containing the highest content of cannabinoid compounds in dried plant material (1.9153% CBD and 0.0681% THC). The hemp extract was prepared by two-phase solvent extraction. First, the plant material was extracted in an organic solvent at 30 °C and concentrated in an evaporator (50 mbar vacuum). Subsequently, the extract was dissolved in a mixture of ethanol and water at 80 °C, then concentrated (50 mbar, vacuum) after ethanol evaporation. The final stage involved decarboxylation (temperature 130 °C), which resulted in an extract containing 215.2 mg/g CBD and 13.3 mg/g THC. 

### 2.3. Preparation of Solutions of Cannabis sativa Extracts

Rapae oleum—ready to use.

Rapae oleum is a mixture of fatty acids: oleic acid (50–67%), linoleic acid (16–30%), linolenic acid (6–14%), palmitic acid (2.5–60%), stearic acid (no more than 3%), eicosenoic acid (5%) and erucic acid (less than 2%) [14].

2.Mixture: Cremophor/ethyl alcohol 96%/NaCl 0.9% in the ratio 1:1:18.

Cremophor is a mixture of polyoxyethylated triglycerides, made by reacting castor oil with ethylene oxide in a molar ratio of 1:35, which acts as a non-ionic surfactant and serves as a formulation vehicle (p.o. and i.v. administration) for poorly-water-soluble pharmacological agents. (Hydrophilic–lipophilic balance (HLB)~12–14.) [15].

To dissolve the extracts, 50 mL of each solution was prepared, 50 mL of Rapae oleum and 50 mL of the mixture: Cremofor—2.5 mL + ethanol 96%—2.5 mL + NaCl 0.9%—45 mL.

The experiment used *Cannabis sativa* extracts with a concentration of 216 mg of pure cannabidiol (CBD) in 1 g of extract. Each rat received a single dose of the extract dissolved in Rapae oleum or Cremophor/ethanol/NaCl mixture at a dose of 30 mg/kg of rat body-weight. Details of individual doses of the extract dissolved in Rapae oleum and in the mixture of Cremophor/ethyl alcohol 96%/NaCl 0.9% are presented in Table 1 and Table 2.

### 2.4. High-Performance Liquid Chromatography 

The extracts were analysed by HPLC [16] on the Accucore C18 column (2.6 μm particle size, 10 cm × 2.1 mm) using the mobile phases of 0.01% formic acid in acetonitrile (A) and 0.01% formic acid (B). After filtration, the samples were diluted in acetonitrile, placed in vials, and automatically injected (1 µL) into the HPLC system (Thermo Scientific, Waltham, MA, USA). The Accucore C18 column temperature was 50 °C and the column was calibrated using standard solutions containing CBDA, CBD, Δ9-THC and Δ9-THCA (concentrations of 2.5 µg/mL, 5 µg/mL and 10 µg/mL were prepared by dilution with acetonitrile). The calibration curve for THC was y = 0.0898x + 0.0541, where R2 = 0.997 and the limit of detection (LOD) was 0.05 µg/mL, and for CBD, y = 0.0889x + 0.0172, where R2 = 0.999 and LOD = 0.008 µg/mL. The gradient was established by starting when 35% A, after 8 min decreasing to 20% A, and 4 min later decreasing to 0% A. The flow rate was 0.300 mL/min and the eluents were analysed using a diode array detector (Thermo Scientific, Waltham, MA, USA) at a wavelength of 230 nm for detection. The data were collected using the software Chromeleon 7.0 [analytical error (n = 10) for Δ9-THCA ± 0.15%, CBDA ± 0.10%, CBD ± 0.5% and Δ9-THC ± 0.13%], and the following cannabinoids were analysed: Δ9-tetrahydrocannabinol (Δ9-THC), Δ9-tetrahydrocannabinolic acid (Δ9-THCA), cannabidiol (CBD) and cannabidiol acid (CBDA).

### 2.5. Animal Study

The study’s protocol was approved by the local Ethics Committee for Animal Research (consent number: Resolution No. 3/2017 of 17 March 2017 of the local Ethics Committee for Animal Research in Poznań), and all experimental procedures were performed as per the Polish regulations for the handling and use of laboratory animals. The experiment was conducted in June 2017 on 48 male Wistar rats (n = 48) weighing 220–250 g; rats were allowed food and water ad libitum and maintained under standard conditions (temperature 20–22 °C; 60–65% relative humidity) on a12-h light/dark cycle (light on at 07:00). The animals were deprived of food 24 h before the start of the experiment and randomly divided into two groups (n = 24), with each group receiving an oral dose of *Cannabis sativa* L. extracts in either (*i*) Rapae oleum or (*ii*) Cremophor EL [Kolliphor® EL, pH-range 6.0–8.0, Sigma Aldrich, St. Louis, MA, USA]/ethanol/saline solution (1:1:18) (later abbreviated as Cremophor), at a dose of 30 mg/kg (cannabidiol dose equivalents). The blood concentrations of CBD and THC were measured at 0.5, 1, 2, 4, 6 and 24 h (4 rats were sacrificed per time point) post-administration by HPLC–MS. 

### 2.6. Analytical Procedure

#### 2.6.1. Blood Sample Preparation

Blood (0.5 mL) was mixed with 10 μL of the internal standard solution (1 µg/mL CBD D3 and THC-D3) and precipitated in five batches, once with 125 μL of frozen ACN. Precipitated blood was centrifuged at 5500 rpm, and then 650 μL of supernatant was transferred to an Eppendorf and precipitated with 200 μL acetone. The sample was frozen at −20 °C for at least 30 min and then centrifuged at −20 °C for 5 min at 1300 rpm. The supernatant was transferred to an HPLC vial and diluted with 700 μL H_2_O. A standard curve was also prepared based on the blood of rats not supplemented with CBD hemp extract.

#### 2.6.2. Brain Sample Preparation

Due to possible discrepancies in the CBD content in parts of the brain, the whole brain was analysed. The brains were weighed and placed in a 5.0 mL Eppendorf before the addition of 10 μL internal standard solution (10 μg/mL of both CBD D3 and THC D3), Magna Lyser Roche LifeScience crushing balls and 0.5 mL of H_2_O UHPLC for the mass spectrometry. The samples were shaken on a vortex at 3.600 rpm for 15 min and the fragmented tissue was extracted with 1.0 mL of frozen acetonitrile UHPLC for the mass spectrometry. The samples were frozen at −20 °C for at least 30 min and then centrifuged at −20 °C for 5 min at 1300 rpm. The supernatant was transferred to an HPLC vial and diluted with 700 μL H_2_O UHPLC for the mass spectrometry. The calibration curve was prepared using 1.5 mL of water instead of the brain, according to the procedure described for the whole blood samples. The content of CBD and THC was calculated per ng/1 g.

#### 2.6.3. LC–MS/MS

The LC–MS/MS system consisted of a CTC Pal liquid sampler, an ABSciex ExionLC column oven, an ABSciex ExionLC system controller, two ABSciex ExionLC high-performance liquid chromatography pumps and a degasser, combined with an ABSciex 4500 QTRAP triple-quadrupole mass spectrometer (MS) equipped with a TurboIon Spray interface and the Analyst software, version 1.6.3. The following chromatographic conditions were applied: Phase A, 0.1% acetate ammonium in H_2_O; and Phase B, ACN:MeOH 1:1, oven temperature 40 °C, Column 100 × 3.0 mm RP C18 Bionacom Velocity, a flow rate of 0.40 mL/min and 20 µL injection volume. The analysis was performed using the gradient method (Table 3). Positive electrospray ionization (ESI) was used to detect the analytes with nitrogen as the curtain gas and nebulizer gas as follows: curtain gas CURŁ 50, IonSpray Voltage IS: 5000 V, temperature 650 °C, Ion Source Gas 1 GS1: 40 and Ion Source Gas 2 GS2: 35. The limit of quantification (LOQ) for both CBD and THC was estimated at 5.0 ng/mL. The calibration curves were obtained in the range of 5–800 ng/mL for CBD and 5–400 ng/mL for THC, with a correlation coefficient of r > 0.995. The analytical procedure was validated and confirmed as suitable for the intended purpose with within-day and between-day coefficients of variation of less than 10%.

#### 2.6.4. Pharmacokinetic (PK) and Statistical Methods

The naive pooling method was used for PK analysis, calculating the mean value of CBD and THC concentration at each sampling time point (based on data from 4 rats). The mean whole-blood concentrations, with standard deviations versus time curves, were plotted, for comparison of profiles, across compounds and formulations. Kinetica 5.1 software was used to assess the following PK parameters of cannabinoids with the non-compartmental analysis applied: *AUC_0–24_*, *C_max_*, *T_max_*, *C_L_*, *MRT*, terminal slope and *V_d_.* Statistical analysis was performed using Student’s *t*-tests, and a *p*-value < 0.05 was considered statistically significant.

## 3. Results

Figure 1 presents the time course of CBD and THC concentrations, and Figure 2 is the time-course of the CBD/THC ratio in whole-blood and brain for the Rapae oleum and Cremophor preparations. There were significantly higher levels (*p* < 0.05) of CBD than THC in both whole-blood and brain, but the CBD/THC concentration ratio decreased with time after administration. The extract dissolved in Rapae oleum initially had lower concentration ratio differences compared to the extract dissolved in Cremophor. The ratio between blood concentration of CBD and THC and brain concentration of CBD and THC for Cremophor initially showed increasing variability, and then, after 6 min, they remained constant, and lower than in the case of Rapae oleum (Figure 2).

Figure 3 shows the ratio of CBD and THC concentrations in Rapae oleum and Cremophor over time intervals. The ratio of concentrations of substances dissolved in the Rapae oleum is higher and more often tends to increase than does that of the substances dissolved in the Cremophor, which reaches a constant level.

The PK data, mean concentrations of THC and CBD in different solvents as well the ratios of *AUC_0–24_* in whole-blood and brain depending on the solvent, are presented in Table 4, Table 5 and Table 6, respectively. The mean concentration (C mean) of CBD and THC in blood and brain were also calculated for every time point and presented in Table 5.

HPLC chromatograms for CBD and THC measurement for whole-blood and brain samples are presented in Figure 4. 

The maximum whole blood concentration (*C_ma_*_x_) and *AUC_0–24_* of CBD (333.62 ng/mL and 1287.65 ng/mL×h, respectively) were about ten times higher than that of THC (26.19 ng/mL and 185.92 ng/mL×h) for the Rapae oleum preparation (Table 4). 

For the Cremophor preparation, the maximum whole-blood concentrations of CBD and THC were 120.13 ng/mL and 18.23 ng/mL, respectively, whereas the whole-blood *AUC_0–24_* was 988.03 and 217.02 ng/mL×h for CBD and THC, respectively (Table 4). The brain bioavailability of CBD was also higher than that of THC, but the difference was less obvious when compared in whole blood. The CBD/THC ratios of A *AUC_0–24_* in the brain were 3.6 and 1.4 for Rapae oleum and Cremophor preparations, respectively (Table 4). 

CBD in whole blood achieved a higher *C_max_* (333.62 ng/mL) and rose later (*T_max_* = 2 h) for the extract dissolved in Rapae oleum, compared to the extract dissolved in Cremophor EL (*C_max_* = 120.13 ng/mL, *T_max_* = 0.5 h). *C_max_* values translate to *AUC_0–24_* values in whole blood, that is, they were higher for the extract dissolved in Rapae oleum (*AUC_0–24_* = 1287.65 ng/mL×h) compared to the extract dissolved in Cremophor EL (*AUC_0–24_* = 988.03 ng/mL×h).

CBD in the brain showed similar values of *C_max_* (*C_max_* = 301.38 ng/mL) and *T_max_* (*T_max_* = 2 h) as in whole blood. Compared to Cremophor (*C_max_* = 274.99 ng/mL, *T_max_* = 0.5 h), these pharmacokinetic values for Rapae oleum were higher and rose later, but in the same time interval as in whole blood (2 h vs. 0.5 h). *C_max_* values translate into A *AUC_0–24_* values in the brain. However, compared to whole blood, in the brain, the *AUC_0–24_* of CBD was significantly higher for the extract dissolved in Rapae oleum (*AUC_0–24_* = 1985.93 ng/mL×h) compared to the extract dissolved in Cremophor EL (*AUC_0–24_* = 943.04 ng/mL×h).

THC in whole blood showed a similar trend in terms of concentrations as CBD, reaching a higher *C_max_* (26.19 ng/mL) and much later (*T_max_* = 4 h) for the extract dissolved in Rapae oleum compared to the extract dissolved in Cremophor EL (*C_max_* = 18.53 ng/mL, *T_max_* = 1 h). However, the *AUC_0–24_* values were comparable in both solvents and even slightly higher for Cremophor (*AUC_0–24_* = 217.02 ng/mL×h) compared to Rapae oleum (*AUC_0–24_* = 185.92 ng/mL×h).

THC in the brain showed significantly higher *C_max_* values in both solvents compared to whole blood. Similar to whole blood, the brain *C_max_* was higher in Rapae oleum than in Cremophor EL (75.76 ng/mL vs. 56.85 ng/mL) but this difference between solvents was not significant. The *T_max_* for THC in both solvents was 4 h. Interestingly, THC dissolved in Cremophor appeared very quickly in the whole blood compared to the brain (*T_max_* = 1 h vs. 4 h, respectively). The brain *AUC_0–24_* values for both solvents were comparable (553.39 ng/mL×h for Rapae oleum and 655.88 ng/mL×h for Cremophor) and even slightly higher for Cremophor. Interestingly, brain A *AUC_0–24_* values were significantly higher than whole blood *AUC_0–24_* values for both solvents.

The *MRT* value of CBD and THC dissolved in Rapae oleum was significantly lower (4.03 and 5.75) than that of Cremophor in whole blood (12.01 and 12.69). In the brain, the *MRT* values for CBD and THC were comparable for both compounds and solvents. Unlike in whole blood, for CBD and THC in the brain, the *MRT* was lower for Cremophor (6.19 and 5.11) than for Rapae oleum (7.76 and 7.19).

The *Vd* (L) in whole blood for CBD was higher for Cremophor (*Vd* = 69.66 L) than for Rapae oleum (*Vd* = 28.29 L); similarly, for THC, the *Vd* for Cremophor was higher than for Rapae oleum (*Vd* = 28.26 vs. *Vd* = 15.15) in whole blood. The brain CBD was similar to that in whole blood; specifically, Cremophor, compared to Rapae oleum, better supported the distribution of CBD to the brain (*Vd* = 51.08 L vs. 30.38 L, respectively). In the case of THC, it was the opposite; namely, Cremophor, compared to Rapae oleum, limited THC distribution in the brain. The volume of distribution in the brain for CBD was significantly higher than for THC in both solvents (Rapae oleum and Cremophor) (Table 4). 

CBD and THC concentrations were determined after rats were given extracts in Rapae oleum and Cremophor after 0.5 h, 1 h, 2 h, 4 h, 6 h and 24 h, showing that CBD and THC concentrations were statistically significantly different in both the brain and the blood and between the two solvents (Rapae oleum and Cremophor) (Table 7).

THC and CBD mean concentrations at different time points in the brain and whole blood in both solvents are presented in Table 7; the values were statistically significant (*p* < 0.05).

## 4. Discussion

For the *Cannabis sativa* extract with a decreased THC content, there were much higher levels of CBD than THC in both the blood and brain, a phenomenon observed for both formulations. However, there were some differences in the bioavailability of both CBD and THC depending on the solvent, with higher CBD concentrations in the whole-blood and brain observed for the Rapae oleum formulation compared to Cremophor, whereas, in the case of THC, the differences were less noticeable. Whether the enhanced CBD concentration in whole-blood administered in Rapae oleum was due to heightened solubility in gastrointestinal fluids or increased drug permeability remains uncertain. The presence of co-solvents (such as ethanol in Cremophor) may indeed modulate gastric and intestinal drug absorption (and thus their pharmacokinetics) depending on the drugs’ concentration and chemistry [17,18]. Indeed, Cremophor may lower the surface tension and improve the dissolution of lipophilic drugs in an aqueous medium by forming micelles to entrap the drugs [19].

The *C_max_* value defines the highest concentration of the substance that was reached and the time required for this level to be reached (*T_max_*). The *AUC_0–24_* value represents the absolute bioavailability of a compound, encompassing the range of concentrations of that substance during its residence in a given tissue or target site. In our study, there was significantly higher total CBD bioavailability in whole-blood and the brain for the Rapae oleum formulation than for Cremophor, which was confirmed by the mean concentrations, *C_max_* and *AUC_0–24_*, with almost all of the CBD dissolved in the Rapae oleum appearing in the brain. Interestingly, in the case of CBD dissolved in Cremophor EL, the maximum concentration in the brain was more than two times higher than in the whole blood. The THC concentration in whole blood showed a similar trend to CBD, and the *AUC_0–24_* values were comparable in both solvents. THC in the brain showed higher *C_max_* values with both solvents compared to in whole blood. Interestingly, THC dissolved in Cremophor appeared very quickly in the whole blood, as compared to in the brain (*T_max_* = 1 h vs. 4 h, respectively), whereas the brain *AUC_0–24_* values for both solvents were comparable but higher than the whole blood’s *AUC_0–24_*. Analysis of the *AUC_0–24_* values for THC in whole-blood and brain shows that absolute bioavailability was higher in the brain than in the blood for both solvents, suggesting the accumulation of CBD and THC in brain tissue. This is supported by clinical observations showing prolonged psychotropic effects after oral administration of THC, CBD and a THC+CBD combination, or *Cannabis sativa* extracts [20]. In addition, the *C_max_* and *AUC_0–24_* values were higher when the extract was dissolved in Rapae oleum than when in Cremophor EL, both for CBD and THC, in whole-blood and the brain, indicating that Cremophor may reduce the absorption of CBD and THC after oral administration. In a study by Bardelmeijer et al., Cremophor retained lipophilic paclitaxel in the lumen of the gastrointestinal tract, possibly by incorporating it into micelles, thereby reducing drug absorption. This may also explain why the absorption of lipophilic CBD and THC encapsulated in Cremophor EL is limited [21]. Deiana et al. conducted Cremophor -based studies to evaluate the CBD pharmacokinetic profile in rats and mice [17]. CBD was extracted from *Cannabis* plants and suspended in Cremophor EL/ethanol/saline in a ratio of 1:1:18. The authors compared whole-blood and brain levels of CBD after oral or intraperitoneal administrations of the drug in Cremophor at 120 mg/kg to rats and mice. Oral dosing resulted in a similar peak for whole-blood concentration in both species (∼2 μg/mL), but the brain peak CBD concentration was six times higher in rats than in mice (8.6 vs. 1.3 μg/g). It was also noted that oral administration of CBD (120 mg/kg) dissolved in the micelle-forming Solutol resulted in enhanced absorption of the drug, compared to a solution based on the emulsion-forming surfactant Cremophor, as evidenced by higher peak concentrations and prolonged exposure in the blood (3.2 μg/mL at 6 h and 2 μg/mL at 2 h) and brain (12.6 μg/mL at 4 h and 8.6 μg/mL at 4 h). In our study, the maximum levels of CBD administered in Cremophor occurred after 0.5 h and were lower than in the study by Deiana et al. (120.13 ng/mL and 274.99 ng/mL for whole-blood and brain, respectively). However, our CBD dose per kg bw was four times lower, with higher CBD bioavailability for the Rapae oleum formulation for whole blood. The authors also evaluated the brain/whole-blood ratio in Cremophor-based studies, reporting a brain/whole-blood ratio of 2.64, similar to that of the present study at 2.29. 

The average residence time of the substance in the body and the rate of elimination of CBD and THC were assessed based on the analysis of *MRT* values. The observed *MRT* of CBD and THC in Rapae oleum was lower (4.03 and 5.75, respectively) than that of Cremophor in whole blood (12.01 and 12.69, respectively). In the brain, *MRT* values showed a similar trend to that of whole blood, but the differences between the two solvents are smaller: 6.19 for CBD and 5.11 for THC in Cremophor, and 7.76 for CBD and 7.19 for THC in Rapae oleum. This suggests that Cremophor prolongs the elimination of CBD and THC from whole blood. However, the analysis of the *Vd* (L) value shows that Cremophor supported the distribution of both these compounds in whole blood, which is consistent with its hydrophilic properties in the whole-blood environment. In the brain, Cremophor supported CBD distribution but limited THC distribution. In addition, the volume of distribution in the brain for CBD was higher than that of THC in both solvents. It is noteworthy that in all cases, the *Vd* values significantly exceeded the weight of the animals (about 400 g), indicating that these substances accumulate in the target tissues.

In vitro studies conducted in human and animal cell models confirm the conversion of CBD to THC in artificial gastric juice [22,23]. It is unclear how adequate levels of Δ9-THC and Δ8-THC can accumulate in humans after oral ingestion of CBD. However, clinical trials in animals and humans provide contradictory evidence for the conversion of CBD to THC in vivo [24]. According to some authors, simulated gastric juice may not sufficiently reflect the conditions in the human body, and Δ9-THC and its metabolites after oral CBD administration should also be detectable in serum [25]. This is supported by a study by Palazzoli et al., who did not detect Δ9-THC in the blood of rats 3 or 6 h after oral administration of CBD at a dose of 50 mg/kg in olive oil [26]. Similar observations were made by Wrey et al., who did not detect Δ9-THC after oral administration of CBD at a dose of 15 mg/kg to mini-pigs [27]. We also did not observe Cremophor to be protective of CBD in the stomach and protect against conversion to THC, as the respective THC-to-CBD ratios in the Rapae oleum and Cremophor remained similar in both whole-blood and brain.

We intended to prepare an extract in which the THC content was reduced, due to its undesirable psychogenic effects. In this approach, the prepared extract is innovative and the presented PK confirm that this goal was achieved. The chemical and pharmacological aspects of THC have been investigated more than those of the non-intoxicating CBD, due to the former’s psychotropic activity and therapeutic properties. However, many papers have reported a wide range of pharmacological activities of CBD in the last few years, encompassing analgesic, anti-inflammatory, antioxidant, antiemetic, antianxiety, antipsychotic and anticonvulsant properties, as well as its cytotoxic effects (exclusively on malignant cell lines) [28]. In view of this published data, the obtained extract is promising, but further PKPD studies are required to examine the dose–response relationship for the different pharmacological effects expected for CBD. In our study, the CBD levels in whole blood were between 1.06–588.03 ng/mL for oil and 1.43–231.43 ng/mL for Cremophor. The oral administration of chocolate cookies to 12 healthy volunteers enriched with a 0.57 mg/kg CBD + 0.28 mg/kg THC mixture resulted in a low whole-blood concentration of about 5 ng/mL for each drug after 1.5–3 h [29]. Similarly low levels in whole blood (range: 0.3–2.6 ng/mL, average 0.93 ng/mL) were noted in 24 volunteers 1 h after oral ingestion of gelatin capsules with *Cannabis sativa* extract containing 0.078 mg/kg CBD + 0.14 mg/kg THC [30]. CBD remained detectable for 3–4 h after administration, resulting in analgesia. In our study, CBD concentrations were detectable up to 24 h after administration and were much higher, with maximum CBD concentrations in the brain of 301.38 ng/mL and 274.99 ng/mL. 

The novelty of our study is that we examined the bioavailability of a new *Cannabis sativa* extract with reduced THC content. Another strength was the modern technology used to quantify the concentration of active substances. Nonetheless, the study was limited by the lack of analysis of the expression of CB1 and CB2 receptors, which would undoubtedly add credibility to results based solely on the concentrations of these substances. 

In conclusion, the oral administration of the *Cannabis sativa* extract with a decreased THC content leads to much higher concentrations of CBD than THC in the whole-blood and brain for both solvent formulations (Rapae oleum and Cremophor). The total bioavailability of both CBD and THC was higher for Rapae oleum than for Cremophor. Since some of the CBD is converted into THC in the body, this should be considered when using *Cannabis sativa* for medical purposes. This THC-reduced hemp extract is a promising candidate for medical applications. 

## Figures and Tables

**Figure 1 nutrients-15-02646-f001:**
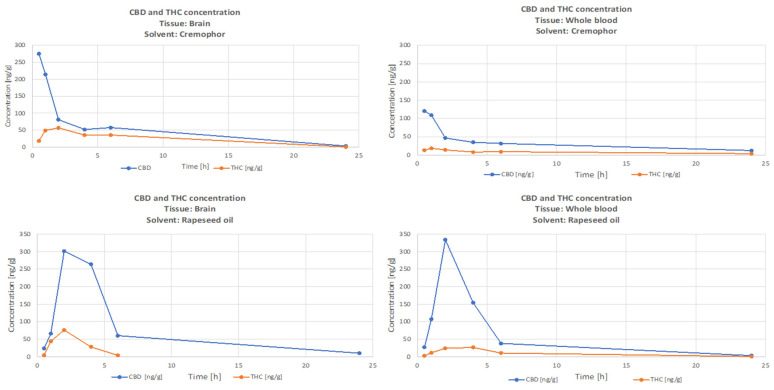
Time course of THC and CBD concentrations in rat whole-blood and brain after a single oral dose of *Cannabis sativa* extract at 30 mg/kg (cannabidiol equivalent dose) in the solvents Cremophor EL [Kolliphor® EL, pH-range 6.0–8.0, Sigma Aldrich]/ethanol/saline solution (1:1:18) formulation and Rapae oleum.

**Figure 2 nutrients-15-02646-f002:**
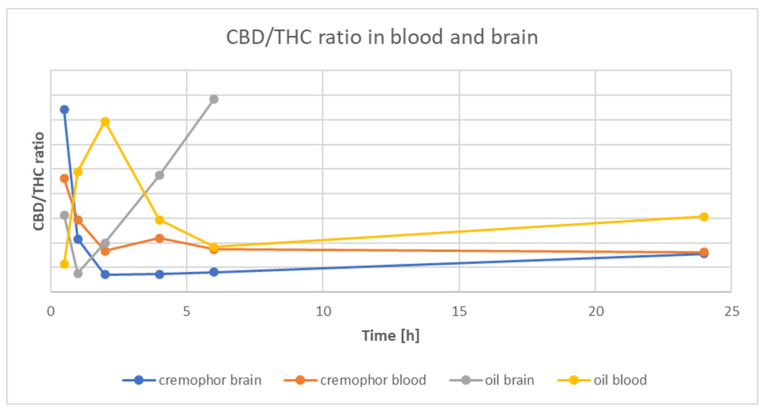
Time course of the CBD/THC ratio in rat whole-blood and brain after a single oral dose of *Cannabis sativa* extract at 30 mg/kg (cannabidiol equivalent dose) in the solvents Cremophor EL [Kolliphor® EL, pH-range 6.0–8.0, Sigma Aldrich]/ethanol/saline solution (1:1:18) formulation and Rapae oleum.

**Figure 3 nutrients-15-02646-f003:**
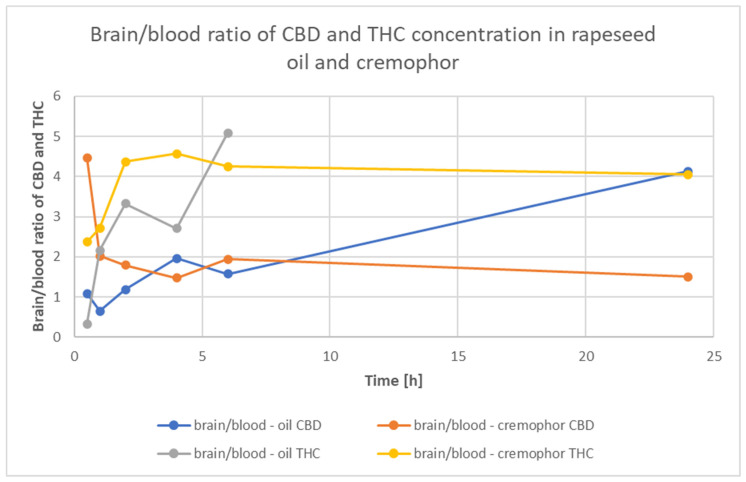
Time course of the brain/blood ratio in rats after a single oral dose of *Cannabis sativa* extract at a 30 mg/kg (cannabidiol equivalent dose) in the solvents Cremophor EL [Kolliphor® EL, pH-range 6.0–8.0, Sigma Aldrich]/ethanol/saline solution (1:1:18) formulation and Rapae oleum.

**Figure 4 nutrients-15-02646-f004:**
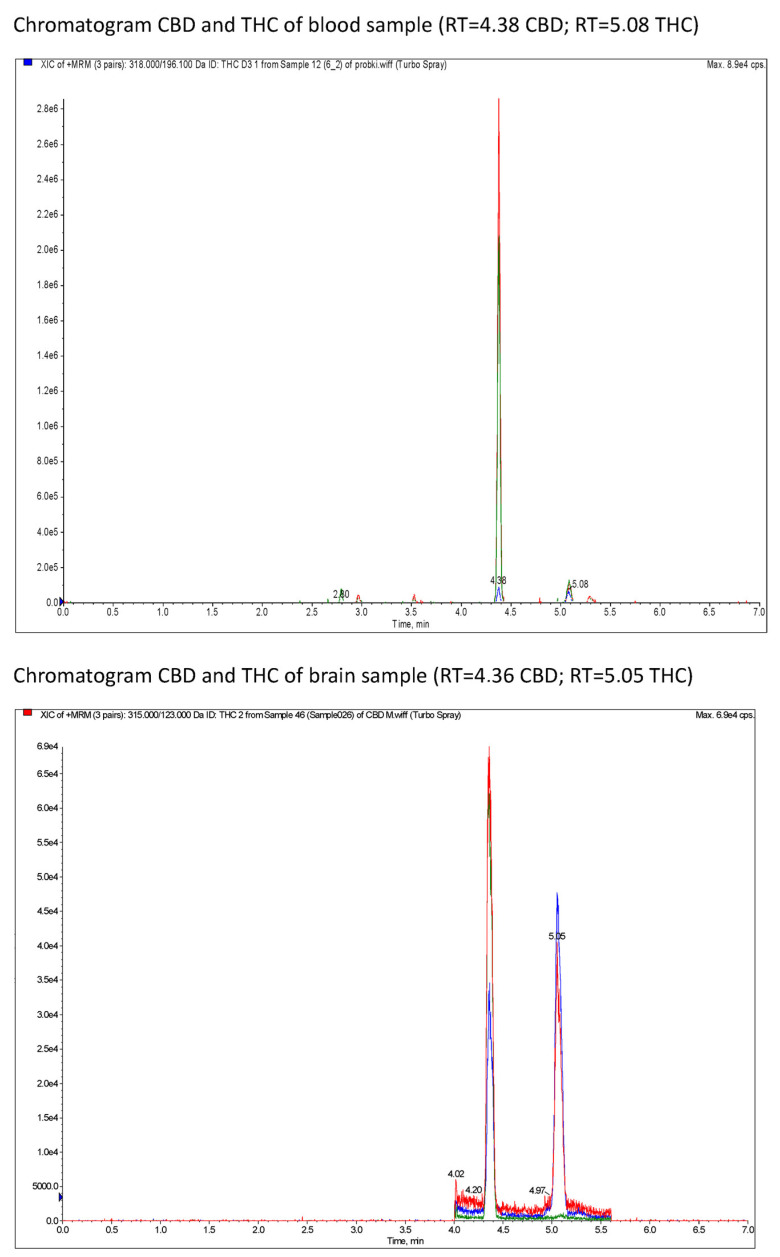
XIC (extracted ion chromatograms) of MRM (multiple reaction monitoring) of CBD and THC of blood samples and brain samples in rats after a single oral dose of CBD and THC via a single oral dose of *Cannabis sativa* extract at 30 mg/kg (cannabidiol equivalent dose) in the solvents Cremophor EL [Kolliphor® EL, pH-range 6.0–8.0, Sigma Aldrich]/ethanol/saline solution (1:1:18) formulation and Rapae oleum.

**Table 1 nutrients-15-02646-t001:** The dose of *Cannabis sativa* extract (mg) dissolved in Rapae oleum and CBD amount (mL) calculated by rat body weight (mg).

Time Point	Rat No. 1	Rat No. 2	Rat No. 3	Rat No. 4
0.5 h	BW 250 g Cse 7.5 mg CBD 0.83 mL	BW 270 g Cse 8.1 mg CBD 0.9 mL	BW 265 g Cse 7.95 mg CBD 0.88 mL	BW 265 g Cse 7.95 mg CBD 0.88 mL
1 h	BW 270 g Cse 8.1 mg CBD 0.9 mL	BW 265 g Cse 7.95 mg CBD 0.88 mL	BW 260 g Cse 7.8 mg CBD 0.87 mL	BW 260 g Cse 7.8 mg CBD 0.87 mL
2 h	BW 260 g Cse 7.8 mg CBD 0.87 mL	BW 260 g Cse 7.8 mg CBD 0.87 mL	BW 250 g Cse 7.5 mg CBD 0.83 mL	BW 250 g Cse 7.5 mg CBD 0.83 mL
4 h	BW 255 g Cse 7.65 mg CBD 0.85 mL	BW 245 g Cse 7.35 mg CBD 0.82 mL	BW 265 g Cse 7.95 mg CBD 0.88 mL	BW 250 g Cse 7.5 mg CBD 0.83 mL
6 h	BW 270 g Cse 8.1 mg CBD 0.9 mL	BW 265 g Cse 7.95 mg CBD 0.88 mL	BW 270 g Cse 8.1 mg CBD 0.9 mL	BW 270 g Cse 8.1 mg CBD 0.9 mL
24 h	BW 250 g Cse 7.5 mg CBD 0.83 mL	BW 245 g Cse 7.35 mg CBD 0.82 mL	BW 270 g Cse 8.1 mg CBD 0.9 mL	BW 240 g Cse 7.2 mg CBD 0.8 mL

BW—body weight; Cse—*Cannabis sativa* extract; CBD—cannabidiol.

**Table 2 nutrients-15-02646-t002:** The dose of *Cannabis sativa* extract dissolved in Cremophor/ethyl alcohol (96%)/NaCl (0.9%) mixture in the ratio 1:1:18, and the CBD amounts calculated by rat body weight.

Time Point	Rat No. 1	Rat No. 2	Rat No. 3	Rat No. 4
0.5 h	BW 255 g Cse 7.65 mg CBD 0.85 mL	BW 270 g Cse 8.1 mg CBD 0.9 mL	BW 255 g Cse 7.65 mg CBD 0.85 mL	BW 260 g Cse 7.8 mg CBD 0.87 mL
1 h	BW 270 g Cse 8.1 mg CBD 0.9 mL	BW 270 g Cse 8.1 mg CBD 0.9 mL	BW 265 g Cse 7.95 mg CBD 0.88 mL	BW 270 g Cse 8.1 mg CBD 0.9 mL
2 h	BW 270 g Cse 8.1 mg CBD 0.9 mL	BW 270 g Cse 8.1 mg CBD 0.9 mL	BW 265 g Cse 7.95 mg CBD 0.88 mL	BW 270 g Cse 8.1 mg CBD 0.9 mL
4 h	BW 270 g Cse 8.1 mg CBD 0.9 mL	BW 270 g Cse 8.1 mg CBD 0.9 mL	BW 270 g Cse 8.1 mg CBD 0.9 mL	BW 270 g Cse 8.1 mg CBD 0.9 mL
6 h	BW 270 g Cse 8.1 mg CBD 0.9 mL	BW 270 g Cse 8.1 mg CBD 0.9 mL	BW 260 g Cse 7.8 mg CBD 0.87 mL	BW 270 g Cse 8.1 mg CBD 0.9 mL
24 h	BW 270 g Cse 8.1 mg CBD 0.9 mL	BW 270 g Cse 8.1 mg CBD 0.9 mL	BW 270 g Cse 8.1 mg CBD 0.9 mL	BW 270 g Cse 8.1 mg CBD 0.9 mL

BW—body weight; Cse—*Cannabis sativa* extract; CBD—cannabidiol.

**Table 3 nutrients-15-02646-t003:** Gradient method of chromatographic conditions to quantify CBD and THC concentrations in the blood and brain.

Time [min]	Phase A	Phase B
0.1	40%	60%
0.5	40%	60%
2.5	2.5%	97.5%
5	2.5%	97.5%
5.5	40%	60%
7	40%	60%

**Table 4 nutrients-15-02646-t004:** Pharmacokinetics of CBD and THC in rat whole-blood and brain after a single oral dose of *Cannabis sativa* extract at 30 mg/kg (cannabidiol equivalent dose) in the solvents Cremophor EL [Kolliphor® EL, pH-range 6.0–8.0, Sigma Aldrich]/ethanol/saline solution (1:1:18) formulation and Rapae oleum.

PK Parameter	CBD		THC	
	Rapae Oleum	Cremophor	Rapae Oleum	Cremophor
	Whoole blood			
*C_max_* (ng/mL)	333.62	120.13	26.19	18.53
*T_max_* (h)	2	0.5	4	1
*C_L_* (L/h)	5.85	7.75	2.56	2.18
*AUC_0–24_* (ng/mL×h)	1287.65	988.03	185.92	217.02
*MRT* (h)	4.03	12.01	5.75	12.69
Terminal slope k_e_ (L/h)	−0.15	−0.11	−0.18	−0.11
*V_d_* (L)	28.29	69.66	15.15	28.26
*AUC*_(CBD)_/*AUC*_(THC)_	6.93	4.55		
	Brain			
*C_max_* (ng/mL)	301.38	274.99	75.76	56.85
*T_max_* (h)	2	0.5	4	4
*C_L_* (L/h)	3.91	8.25	0.87	0.73
*AUC_0–24_* (ng/mL×h)	1985.93	943.04	553.39	655.88
*MRT* (h)	7.76	6.19	7.19	5.11
Terminal slope k_e_ (L/h)	−0.10	−0.15	−0.11	−0.20
*V_d_* (L)	30.38	51.08	6.17	3.73
*AUC_(CBD)_/AUC* _(THC)_	3.59	1.44		

**Table 5 nutrients-15-02646-t005:** Mean CBD and THC concentrations [ng/g] in rat blood and brain after a single oral dose of *Cannabis sativa* extract at 30 mg/kg (cannabidiol equivalent dose) in the solvents Cremophor EL [Kolliphor® EL, pH-range 6.0–8.0, Sigma Aldrich]/ethanol/saline solution (1:1:18) formulation and Rapae oleum.

Time [h]	Mean Concentration [ng/g] ± SD for Rapae Oleum	Mean Concentration [ng/g] for Cremophor	*p*-Value
CBD blood			
0.5	26.905 ± 21.476	120.134 ± 93.505	0.036
1	106.913 ± 51.028	108.887 ± 35.391	0.006
2	333.619 ± 138.971	46.538 ± 15.707	0.041
4	153.554 ± 145.61	35.125 ± 4.710	0.036
6	37.584 ± 11.297	31.416 ± 9.282	0.006
24	2.759 ± 2.120	12.315 ± 18.045	0.036
CBD brain			
0.5	23.327+/6.725	274.994 ± 103.691	0.045
1	65.391+/27.232	213.491 ± 66.081	0.031
2	301.388+/155.739	81.114 ± 23.387	0.033
4	263.467+/228.678	52.119 ± 12.179	0.038
6	59.650+/32.649	57.715 ± 25.882	0.001
24	9.797+/7.354	3.604 ± 0.493	0.028
THC blood			
0.5	11.885 ± 18.039	13.05 ± 10.870	0.003
1	10.951 ± 5.213	18.535 ± 1.984	0.016
2	24.029 ± 7.225	13.988 ± 3.297	0.016
4	26.198 ± 19.106	8.008 ± 1.480	0.031
6	10.259 ± 2.095	9.045 ± 4.386	0.004
24	0.445 ± 0.210	3.788 ± 4.592	0.043
THC brain			
0.5	3.719 ± 2.345	18.575 ± 11.626	0.037
1	43.592 ± 24.786	49.521 ± 9.469	0.004
2	75.768 ± 43.807	56.85 ± 7.386	0.009
4	27.792 ± 15.206	35.976 ± 7.445	0.008
6	3.7605 ± 2.612	35.947 ± 12.691	0.043
24	<LOQ*	1.154 ± 0.577	0.05

LOQ*—limit of quantification = 5 ng/mL.

**Table 6 nutrients-15-02646-t006:** *AUC_0–24_* ratio of CBD and THC in rat whole-blood and brain after a single oral dose of *Cannabis sativa* extract at 30 mg/kg (cannabidiol equivalent dose) in the solvents Cremophor EL [Kolliphor® EL, pH-range 6.0–8.0, Sigma Aldrich]/ethanol/saline solution (1:1:18) formulation and Rapae oleum.

**CBD**	Rapae oleum	*AUC_brain_:AUC _whole_* _blood_	1.54
	Cremophor	*AUC_brain_:AUC _whole_* _blood_	0.95
**THC**	Rapae oleum	*AUC_brain_:AUC _whole_* _blood_	3.72
	Cremophor	*AUC_brain_:AUC _whole_* _blood_	1.22

**Table 7 nutrients-15-02646-t007:** THC and CBD mean concentration over time in rat brain and whole blood after a single oral dose of *Cannabis sativa* extract at 30 mg/kg (cannabidiol equivalent dose) in the solvents Cremophor EL [Kolliphor® EL, pH-range 6.0 – 8.0, Sigma Aldrich]/ethanol/saline solution (1:1:18) formulation and Rapae oleum.

Tissue/Rapae Oleum			*p*-Value	Tissue/Cremophor			*p*-Value
Blood/Rapae Oleum	THC	CBD		Blood/Cremophor	THC	CBD	
0.5	11.885	26.905	0.024	0.5	13.05	120.133	0.043
1	10.9508	106.9125	0.044	1	18.535	108.887	0.039
2	24.029	333.619	0.045	2	13.988	46.538	0.031
4	26.1975	153.554	0.039	4	8.008	35.125	0.036
6	10.259	37.584	0.033	6	9.045	31.416	0.032
24	0.445	2.759	0.040	24	3.788	12.315	0.031
Brain/Rapae oleum	THC	CBD		Brain/Cremophor	THC	CBD	
0.5	3.719	23.327	0.040	0.5	18.575	274.994	0.046
1	43.593	65.391	0.013	1	49.521	213.491	0.035
2	75.768	301.388	0.034	2	56.85	81.1135	0.011
4	27.792	263.466	0.043	4	35.976	52.1185	0.011
6	3.7605	59.650	0.046	6	35.947	57.7145	0.014
24	0	9.797	0.05	24	1.154	3.604	0.030

## Data Availability

Not applicable.

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
