# Peer review of "THC-Reduced Cannabis sativa L.—How Does the Solvent Determine the Bioavailability of Cannabinoids Given Orally?"

_nutrients, 2023, doi:10.3390/nu15122646_

Round 1
Reviewer 1 Report
The manuscript deals with industrial hemp extract with reduced content of THC which is given to experimental animals. Pharmacokinetic studies indicate a typical body distribution of THC and CBD. The last one reached higher concentrations in serum and brain tissue as expected. Some insignificant conversion of CBD to THC was detected by the authors. Additionally they studied the influence of lipophilic solvents such as cremophor and rapeseed oil.
What are the psychotropic effects of the converted into THC CBD remains unclear and needs further elucidation. Human studies clearly indicated that CBD does not have psychotropic effects at all, but the use of solvents may change this.
The manuscript as whole represents a nice and precisely performed pharmacokinetic study which is interesting enough to be published.
Minor change requested:
r. 289 "DIsCUSSION" -> DISCUSSION
Author Response
Dear Reviewer,
we would like to thank you for taking the time and effort to review our manuscript. We appreciate your thorough analysis of our article. We have taken note of your comment and made the correction you suggested.

Reviewer 2 Report
The manuscript submitted to the nutrients deals with an important topic. CBD and other non-hallucinogenic cannabinoids, together with ‘designer drugs’ are hot topic in pharmaceutical sciences and toxicology. Wider knowledge about the pharmacokinetics of CBD and THC might contribute to better understanding the ADMET of other cannabinoids and its derivatives, as well.
The present study has been designed very well, and the manuscript is also written in good style. I will strongly recommend the acceptance of this paper after clarification several questions.
1. In the abstract and the manuscript “THC-reduced hemp extract” is mentioned consequently. Although, the extraction method is correctly and accurately described, the chemical composition of oil and Cremophor extracts are not discussed. In Methods section (between lines #86 and #87) a table is inserted with values of CBD-A, CBD, d9-THC-A and d9-THC. For which extract are these values given for?
Could you provide information about CBD and THC content of the extracts for oil and Cremphore extracts, respectively? 1. Could you give a short explanation what do you consider under “THC-reduce” adjective?
I do understand, that hemp variety KC Dora has only one tenth of the Cannabis indica; however, I do not see in the manuscript how selective extraction for reduced THC was achieved.
2. Could you provide HPLC chromatograms for CBD and THC measurement for whole blood and brain samples?
3. It would be also helpful to see the LOD with unites in lines #109-110. Please add the LOQ values, as well.
4. Sex and weight of the Whistar rats are not given in the Methods section. Please add this information.
5. Please provide further details about protocol approvement (year, number of decision).
Check and correct your manuscript for plant and drug names:
· Cannabis sativa L. (L is not italicized; only ‘C’ is capital letter, ‘s’ is not)
· Use the current method for drug name rapeseed oil in Latin, which is ‘Rapae oleum’.
Check the text for typos, for example:
- use decimal points instead of decimal commas (for instance: table on page #2, Table 2 on page #6
- remove the highlighted background in Table 1
- make sure to insert space between number and unit (for instance: line #356)
- make sure to remove space in following cases 6.0 – 8.0 (line #207); use endash for all ranges (5–800 – line #165)
- use lower case for H2O
- use symbol ± instead of +/- in Table 3
Proofread by native speaker would significantly improve the quality of present version.
Author Response
Dear Reviewer,
Thank you for the valuable and thoughtful comments. All comments have been carefully considered and addressed in the Point-to-Point response. Any changes made in the manuscript are highlighted in yellow. The Proofreading Service UK has corrected the manuscript.

Round 2
Reviewer 2 Report
Thank you very much for your answers. All my questions were answered.
There are still minor errors/typos in the text:
- Cannabis Ssativa in the title
- Rapae oleum is not necessary to be italicized
- Insert space between value and unit (e.g. Table 1. 250g should be 250 g or 0.82ml should be 0.82 ml). Check the whole text for similar typos.
- references in lines #132-133, #139 should be inserted according to the journal guideline
Author Response
Dear Reviewer,
thank you very much for noticing a few more minor errors. We improved them as follows, based on your suggestions.
Please find attached the text of the corrected publication and the point-to-point answer.
Kind regards,
Joanna Bartkowiak-Wieczorek
